# Annealing of Boron-Doped Hydrogenated Crystalline Silicon Grown at Low Temperature by PECVD

**DOI:** 10.3390/ma12223795

**Published:** 2019-11-19

**Authors:** Marta Chrostowski, José Alvarez, Alessia Le Donne, Simona Binetti, Pere Roca i Cabarrocas

**Affiliations:** 1TOTAL S.A., 2, Place Jean Millier-La Défense 6, 92069 Courbevoie CEDEX, France; 2LPICM-CNRS, Ecole polytechnique, Institut Polytechnique de Paris, 91128 Palaiseau, France; pere.roca@polytechnique.edu; 3Institut Photovoltaïque d’Ile-de-France (IPVF), 18 boulevard Thomas Gobert, 91120 Palaiseau, France; jose.alvarez@centralesupelec.fr; 4GeePs, CNRS UMR 8507, CentraleSupélec, Université Paris-Sud, Université Paris-Saclay, Sorbonne Université, 11 rue Joliot-Curie, Plateau de Moulon, F-91192 Gif-sur-Yvette CEDEX, France; 5Dept. of Materials Science and Milano-Bicocca Solar Energy Research Center (MIB-SOLAR), University of Milano-Bicocca, via Cozzi 55, I-20125 Milano, Italy; alessia.ledonne1@unimib.it (A.L.D.); simona.binetti@unimib.it (S.B.)

**Keywords:** silicon epitaxy, PECVD, B–H complexes

## Abstract

We investigate low-temperature (<200 °C) plasma-enhanced chemical vapor deposition (PECVD) for the formation of p–n junctions. Compared to the standard diffusion or implantation processes, silicon growth at low temperature by PECVD ensures a lower thermal budget and a better control of the doping profile. We previously demonstrated the successful growth of boron-doped epitaxial silicon layers (p+ epi-Si) at 180 °C. In this paper, we study the activation of boron during annealing via dark conductivity measurements of p+ epi-Si layers grown on silicon-on-insulator (SOI) substrates. Secondary Ion Mass Spectroscopy (SIMS) profiles of the samples, carried out to analyze the elemental composition of the p+ epi-Si layers, showed a high concentration of impurities. Finally, we have characterized the p+ epi-Si layers by low-temperature photoluminescence (PL). Results revealed the presence of a broad defect band around 0.9 eV. In addition, we observed an evolution of the PL spectrum of the sample annealed at 200 °C, suggesting that additional defects might appear upon annealing.

## 1. Introduction

Global energy consumption is constantly growing and expected to increase by nearly 50% by 2050 [1]. In this context, the need for a more varied energy mix and low-carbon emitting sources is crucial to face the climate situation. Photovoltaic (PV) energy has had one of the fastest growth rates in renewables generation and already is the cheapest form of electricity in many countries [1,2]. In 2019, the cumulated installed PV capacity exceeded 500 GW, led by Al BSF and PERC solar modules, which are both based on p-type silicon wafers. However, in recent years part of the industry has been shifting to n-type wafers and architectures, such as the n-PERT or i-TOPCon cells, with 23.2% and 24.58% record efficiencies, respectively [3,4]. While boron diffusion or implantation are the standard junction formation techniques, they require high processing and/or annealing temperatures, thus increasing the overall thermal budget of the fabrication and the risk of wafer damage, such as impurities diffusion and thermal donor activation [5]. This paper aims at proposing an alternative method for the formation of the junction, consisting of growing an epitaxial boron emitter (p+ epi-Si) by low-temperature (<200 °C) plasma-enhanced chemical vapor deposition (PECVD). The advantages of such technique are the lower thermal budget, the simplification of the process flow by reducing the number of fabrication steps [6] and the precise control of the doping profile. Previous studies have shown the potential of low-temperature PECVD for solar cells applications. Labrune et al. achieved a 14.2% device formed by the homoepitaxial growth of boron-doped layers on an n-type crystalline silicon wafer, from a H_2_/SiH_4_/B_2_H_6_ gas mixture [7]. Leal et al. demonstrated excellent structural properties for highly boron-doped epi-Si layers deposited at 175 °C using a SiF_4_/H_2_/Ar gas mixture, supported by HR-TEM imaging and X-ray diffraction [6].

In this paper, we focus on the impact of annealing on the properties of p+ epi-Si layers grown at 180 °C. First, the structure of the layers is assessed using spectroscopic ellipsometry. The electrical properties are investigated via coplanar dark conductivity measurements as a function of temperature. It is shown that annealing at temperatures above that of the substrate during growth is needed to significantly increase the conductivity. The results are confirmed by Hall effect measurements performed in the as-grown state and after annealing. We also examine the elemental composition using SIMS and characterize the layers by low-temperature photoluminescence—a widely used and recognized technique for the study of defects and impurities in silicon [8].

## 2. Materials and Methods

### 2.1. Doped epi-Si Growth by PECVD at Low Temperature

The p+ epi-Si layers were grown in the Octopus II from Indeotec—a semi-industrial capacitively coupled radio-frequency PECVD reactor (13.56 MHz) at a process temperature of 180 °C. Phosphorus-doped (100) double-side polished FZ crystalline silicon wafers from Topsil with a resistivity of 1–5 Ω.cm were used as substrates. Prior to loading into the reactor, the wafers were dipped into a 5% diluted HF bath to remove the native oxide and obtain a hydrogenated surface that is necessary for a high-quality epitaxial growth. The p+ epi-Si layers were grown from a gas mixture consisting of H_2_/SiH_4_ as the silicon gas source and (CH_3_)_3_B (trimethylboron: TMB) was added as p-type precursor. The gas mass flow rates were set to 500, 35, and 1 sccm, respectively. TMB was chosen as dopant gas because thermal decomposition of B_2_H_6_ has been demonstrated to be a cause of cross-contamination in PECVD reactors [9]. The process pressure was set to 2 mbar and the RF power to 50 W, the electrode area being 350 × 450 mm^2^. These parameters resulted in a growth rate of 1.8 ± 0.1 Å/s. The epi-Si layer thickness was varied in the range of 200 to 1000 nm to be suited to the various characterization techniques.

### 2.2. Characterization Methods

The layers thickness, crystalline quality and surface state (roughness, oxidation) were assessed using ex-situ spectroscopic ellipsometry with a phase modulated ellipsometer—the Uvisel2 from Horiba Jobin-Yvon. The measurements were performed in the range 0.5 to 5.5 eV using a Xe source at an incident angle of 75°. The elemental composition of the layers was deduced from SIMS measurements carried out with a Cameca IMS 7f spectrometer. For these measurements, we chose a highly boron-doped (100) c-Si wafer (resistivity <0.005 Ω cm) as substrate. Samples with p+ epi-Si layers grown on silicon-on-insulator (SOI) substrates were prepared for dark conductivity measurements. Coplanar Al contacts were evaporated through a shadow mask (5 mm long with a 2 mm gap) and their ohmic behavior was checked before the measurements. The conductivity measurements were carried out under vacuum and the temperature was varied from room temperature (RT) to 400 °C. SOI samples were also used to perform Hall effect measurements to assess the carrier mobility and concentration. Finally, photoluminescence (PL) spectra in the 1000–1700 nm spectral range were collected at 14 K. Thicker layers (1 µm) were grown for this measurement to maximize the response from the epi-Si films. All PL measurements were performed with a spectral resolution of 6.6 nm using a standard lock-in technique in conjunction with a single grating monochromator and a short wavelength enhanced InGaAs detector with maximum responsivity at 1540 nm. A quantum well laser (λ_exc_ = 805 nm) with a power density of 10.7 W/cm^2^ was used as excitation source. A cooling system consisting of rotary pump, turbomolecular pump, and He closed circuit cryostat was used to perform PL measurements at low temperature.

## 3. Results and Discussion

### 3.1. Structural and Elemental Analysis

Figure 1a shows the imaginary part of the pseudo-dielectric function of p+ epi-Si layers, as-grown and after annealing at temperatures between 250 and 400 °C. All spectra have the two characteristic peaks E_1_ and E_2_ of crystalline silicon at 3.4 and 4.2 eV respectively, demonstrating that the PECVD process resulted in a crystalline growth. The difference in intensity between the c-Si reference spectrum and that of the epi-Si (Figure 1b) is associated with the presence of surface roughness and oxidation as well as to a decrease in the crystalline quality, caused by the impurities incorporation in the epi-Si films, as will be described in the next paragraph. There is also an evolution of the spectra of the different epi-Si layers as function of the annealing temperature, as reported in the two zoomed-in graphs (Figure 1b,c), and summarized in Figure 1d. First, in Figure 1b there is an increase in the intensity of the E_1_ peak when increasing the annealing temperature, demonstrating a change in the structure of the films and more precisely an improvement in the crystalline quality. Secondly, we notice a shift towards the lower energies of the annealed samples as compared to that of the as-grown and of the c-Si reference, which is related to a change in the roughness of the layers. Finally, the interference fringes at low energies reveal the presence of a defective interface and can also be used to estimate the thickness of the films (Figure 1c) via modelling and fitting with the Bruggeman effective medium approximation. Here the studied layers are around 200 nm thick.

SIMS measurements were performed to analyze the elemental composition of the p+ epi-Si layers (Figure 2). Boron in the epi-Si layer has a flat profile—characteristic of the low-temperature epitaxial growth—and a concentration of 3 × 10^19^ at/cm^3^, slightly higher than that of the highly doped substrate. The hydrogen concentration of 2 × 10^21^ at/cm^3^ might seem very high for a crystalline material; however a significant hydrogen concentration has been previously reported to be correlated with a tetragonal lattice and its impact on the structure of epi-Si layers was also described [10]. In this study, the focus is on the electrical properties of the layer and, in particular, on the activation of the dopants. The high hydrogen concentration in the doped epi-Si layers most probably leads to the formation of the well-known B–H complexes and thus to the neutralization of the substitutional boron atoms [11,12]. Upon annealing at temperatures above 200 °C hydrogen desorbs from the layer [13] resulting in the deactivation of the B–H complexes and thus in an increase in the conductivity and carrier concentration, as will be discussed below. As previously reported in the literature, we also notice hydrogen diffusion into the substrate [14]. The oxygen peak at the interface between the epi-Si layer and the substrate is attributed to a partial regrowth of native oxide after the HF dip and before loading the sample into the reactor and pumping down. Finally, it is worth noticing that carbon and oxygen concentrations in the layer are at a level comparable to that of boron. Carbon comes from the dopant precursor (TMB) and its role on the structural and/or electrical properties has not been studied yet. On the other hand, there is some outgassing in the reactor as well as contamination brought in by the substrate and its carrying plate, which leads to an oxygen profile decreasing from the substrate. In crystalline silicon oxygen can act as a donor thus reducing the acceptor density.

It has been shown by ellipsometry that annealing influences the structure of the epi-Si layers. SIMS profiling revealed high-impurity concentrations in the as-grown state. The next paragraph will discuss how the observed change in the structure affects the electrical properties of the p+ epi-Si films.

### 3.2. Impact of Annealing on the Electrical Properties

We have previously demonstrated that annealing at 300 °C allows activation of the dopants in the p+ epi-Si layers and to achieve a doping efficiency of around 50% [13]. Here, we study the activation of the dopants by measuring the dark conductivity as function of the temperature. Three temperature cycles were performed as described in the legend of Figure 3. First, we can see that before annealing, the dark conductivity of the as-grown p+ epi-Si is very low for a doped crystalline silicon film, its value of 3.10^−4^ S/cm being comparable to that of a microcrystalline material in its as-deposited state [15]. The temperature increase during cycle 1 shows two regimes, where the change in the slope takes place around the p+ epi-Si growth temperature (180 °C). After cooling down to RT, the conductivity of the layer has increased by over three orders of magnitude and the activation energy (E_a_) has decreased from 0.12 eV to 0.024 eV. During cycle 2, an increase in conductivity is observed when the temperature goes beyond 250 °C and reaches a plateau with an E_a_ of 0.002 eV. A small raise in conductivity is seen for cycle 3; however, it is much less significant. When the sample is cooled down from 450 °C to RT the activation energy is negative, which is characteristic of a metallic material. Such a behavior could be associated with an increase of defects in the layer or to a diffusion of Al. These measurements, in particular the first cycle, and the high H concentration measured by SIMS confirm the hypothesis that an increase in conductivity is achieved by dissociating the B–H complexes thus activating the boron atoms in the layer. It is interesting to note that the sharpest increase in conductivity occurs when the temperature goes above that during the growth process.

Hall effect experiments have been performed on SOI samples annealed at 200 and 300 °C in forming gas (Table 1). First, one can see that in the case of the as-grown samples, the carrier concentration is much lower than expected for an emitter, but more importantly the majority carriers measured are electrons (n-type). This corroborates the fact that the dopants are not active in the as-grown p+ epi-Si. We presume that because our epitaxial layer has a low conductivity, the current flows preferably through the substrate, i.e., the top layer of the SOI wafer, which is a lowly doped 700 nm n-type c-Si. After the first annealing at 200 °C, an increase by three orders of magnitude in the carriers concentration together with a change in the sign of the Hall voltage, i.e., of the carrier type, are observed, indicating the activation of the dopants in the p+ epi-Si layer. A decrease of the mobility from 677 to 41 cm^2^/Vs is also noted, as expected for a highly doped layer. A subsequent annealing at 300 °C leads to a further increase in the carrier concentration and a decrease in mobility up to 19 cm^2^/Vs. Such low mobility values compared to theory (52 cm^2^/Vs for a carrier concentration of 6.3 × 10^19^ at/cm^3^ [16]) could imply an onset of the layer degradation. It is surprising that the carrier concentration after the 300 °C annealing is twice as high as the boron concentration measured by SIMS. Such discrepancy might be explained by the fact that the layers are grown on different substrates (FZ c-Si and SOI) or more likely by an overestimation of the carriers by the Hall effect system (systematic error that we have observed). As a matter of fact, if we consider the value of the conductivity after cycle 2 (Figure 3) and the value of the mobility from Hall effect after annealing at 300 °C (Table 1), by a simple calculation we get an electron concentration of 1.9 × 10^19^ at/cm^3^.

### 3.3. Defect Characterization by Low-Temperature PL

Finally, the low carrier mobility and the high impurities concentration in the epi-Si films led to the sample characterization by low-temperature PL, to assess the presence of defects in the layers. PL measurements of as-grown and annealed p+ epi-Si samples are shown in Figure 4. Both spectra present the free exciton (FE) emission typical of c-Si at 1.1 eV, along with its phonon replicas, and a broad band centered around 0.9 eV, associated with a defect related recombination center in the epitaxial layers. However, in the case of the p+ epi-Si sample annealed at 200 °C, a shift to lower energies and an increase in intensity of the band at 0.9 eV were observed. Correspondingly, an intensity decrease of the FE peak occurs, since the new recombination path associated with the emission at 0.9 eV becomes a competitive channel for the carrier recombination.

Complementary PL measurements were performed on intrinsic epi-Si layers (grown at 180 °C) before and after annealing. First, we observe that the FE peak at 1.1 eV is twice less intense than that of the doped layers, maybe due to some non-radiative recombination channels active in the intrinsic epi-Si layers and related to a lower hydrogen concentration (confirmed by SIMS results not shown here). In this case, no significant evolution of the spectrum is observed after annealing. Moreover, the broad band at 0.9 eV is very similar to that of the annealed p+ epi-Si, which suggests that the defects in the epi-Si layers are neither B nor C related, as these elements are not present in the intrinsic layers. A literature review of plasma treatment of crystalline silicon [17,18,19] indicates that the broad peak around 0.9 eV might be related to hydrogen, consistently with the high hydrogen content in the epi-Si layers. In particular, Weman et al. suggest that this broad band arises from strain around extended defects such as hydrogen platelets. HRTEM (high-resolution transmission electron microscopy) characterization will be carried out to investigate these defects.

## 4. Conclusions

We have described the change in conductivity during the annealing of p+ epi-Si layers grown by PECVD. It has been shown that the highest increase in conductivity occurs upon annealing at temperatures above that during growth, i.e., 180 °C. After three temperature cycles, the conductivity of the layer has increased by over four orders of magnitude. Hall effect measurements confirm the activation of dopants after annealing and suggest a structural degradation leading to low mobility values (20 cm^2^/Vs). In addition, SIMS profiles reveal a very high hydrogen concentration, supporting the hypothesis of B-H complexes formation and thus the neutralization of substitutional boron in the as-grown state. Finally, low-temperature PL experiments point in the direction of hydrogen related defects, such as platelets.

## Figures and Tables

**Figure 1 materials-12-03795-f001:**
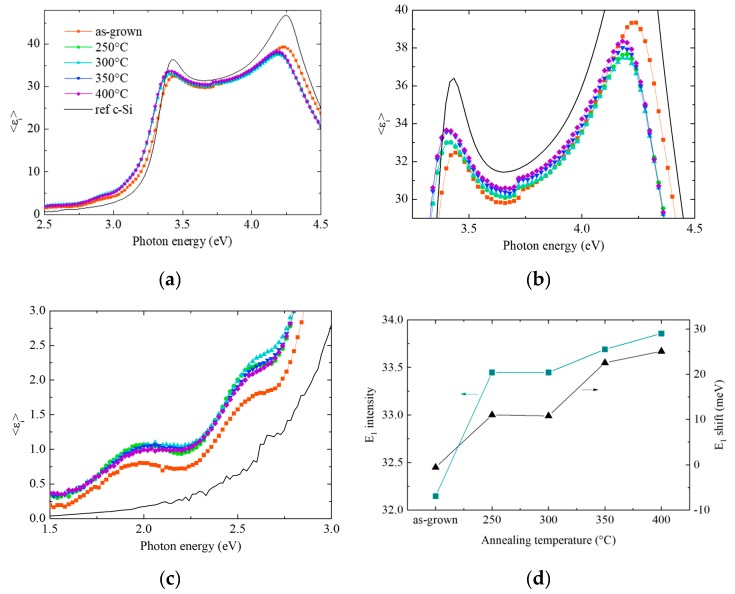
(**a**) Imaginary part of the pseudo-dielectric function of epi-Si layers as-grown and after annealing at temperatures from 250 to 400 °C, and of a c-Si wafer as reference (black line) (**b**) Zoom-in on the two c-Si characteristic peaks E_1_ and E_2_ (**c**) Zoom-in on the interference fringes and (**d**) E_1_ peak intensity (left) and shift (right) as compared to the position of the c-Si reference as functions of the annealing temperature.

**Figure 2 materials-12-03795-f002:**
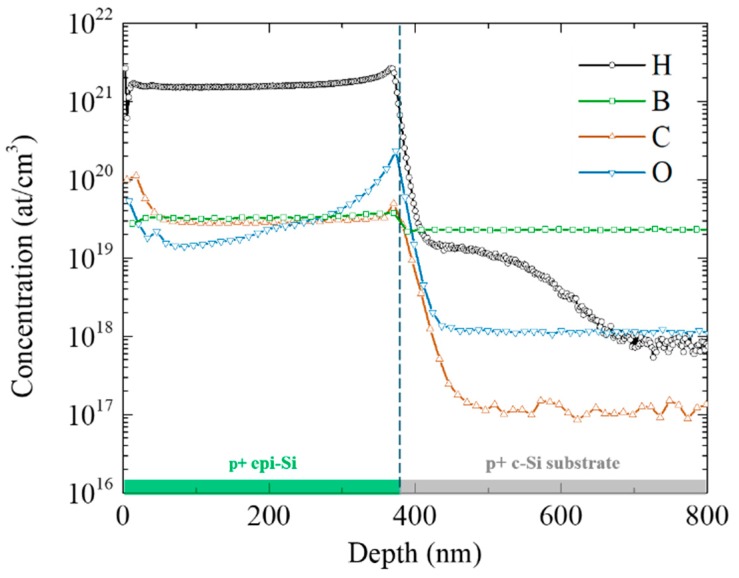
SIMS profile of an as-grown p+ epi-Si layer deposited on a highly boron-doped (100) FZ c-Si substrate. The dashed line represents the interface between the epi-Si layer and the substrate.

**Figure 3 materials-12-03795-f003:**
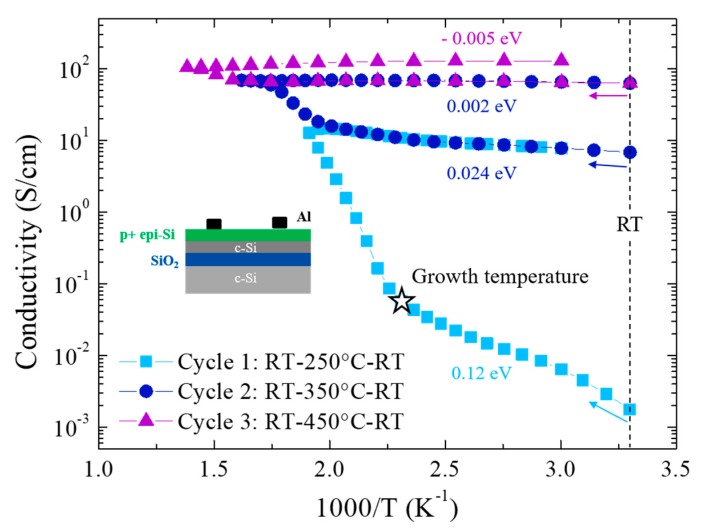
Three cycles of dark conductivity measurements as function of the temperature were performed on SOI samples. The star corresponds to the growth temperature (180 °C).

**Figure 4 materials-12-03795-f004:**
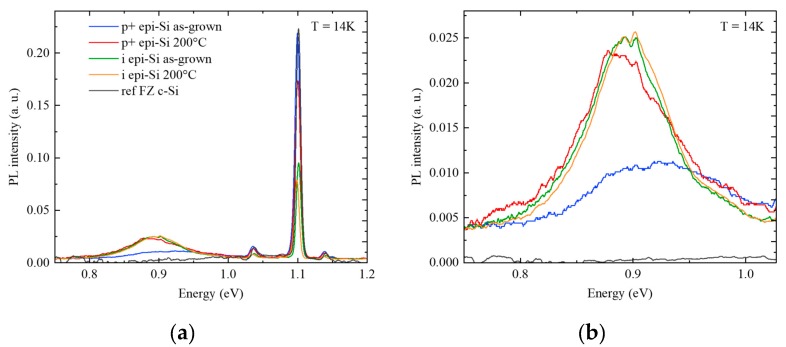
(**a**) PL at 14K before and after annealing p+ and intrinsic epi-Si layers. The spectrum of the c-Si substrate is added as a reference. (**b**) Zoom-in on the broad band at 0.9 eV.

**Table 1 materials-12-03795-t001:** Carriers type, concentration and mobility extracted from Hall effect measurements before and after annealing of the p+ epi-Si layers.

Carriers Properties	As-Grown	200 °C	300 °C
Type	N	P	P
Concentration (at/cm^3^)	5.2 × 10^15^	7.6 × 10^18^	6.3 × 10^19^
Mobility (cm^2^/Vs)	677	41	19

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
