# Peer review of "Annealing of Boron-Doped Hydrogenated Crystalline Silicon Grown at Low Temperature by PECVD"

_materials, 2019, doi:10.3390/ma12223795_

Round 1

Reviewer 1 Report

See attached file

Reviewer 2 Report

The authors investigated properties of B-doped Si epitaxial films grown by low temperature plasma-enhanced chemical vapor deposition (PECVD) aiming at Si p-n junction formation with a lower thermal budget. The measurement data are interesting, useful and original. This shows high potential of Si epitaxial films grown by PECVD for future device fabrication process. While some local points had better to be revised to improve the quality as follows, I can suggest this manuscript for publication after minor revision.

(1) [Page 2, Line 21] The authors should check the value of "p+ epi-Si layers" thickness, because it varied from 250 nm through this manuscript (e.g. 370 nm for SIMS and 1000 nm for PL).

(2) [Page 2, Lines 21-23] The authors had better to add a brief comment (or purpose) of TMB use, because it could become an origin of carbon incorporation in the plasma CVD.

(3) [Page 2, Lines 21-24] The authors had better to add some process conditions such as partial pressures of SiH4, TMB and H2 as well as electrode area for plasma generation. Moreover, showing references for this experimental condition will help readers.

(4) [Page 4, Lines 1-18] The authors had better to comment on the fact that concentrations of carbon and oxygen are at a comparable level as that of boron. For example, this will give a possibility that oxygen atoms act as a donor and reduce hole concentration.

(5) [Page 5, Lines 4-6] Concerning "negative activation energy, characteristic of a metallic material, ... associated to an increase of defects", the authors can think simply that the characteristics are the nature of a heavily doped (degenerate) semiconductors.

(6) [Page 5, Lines 13-25] & Table 1] Concerning the carrier concentration of 6.3x1019 /cc for 300 oC, it is strange that the value far exceeds the boron concentration obtained by SIMS. The authors had better to comment on some origins of such a large disagreement. While there might be a large error in SIMS determination, current density in Hall measurement might have a large dependence on depth from surface due to increasing tendency of oxgen concentration (if it works as a donor) and might give a large error.

(7) [Page 6, Fig. 4] The authors had better to revise PL spectra for readers to see more details. Height and its order of PL peaks around 1.1 eV seem also important and worth to be discussed.

(8) [Page 6, Line 18] In general, "spectroscopy" for HRTEM should be "microscope".
